# Characterization of the Domestic Cat Population of Uruguay: Breeds, Coat Colors, Hair Length, Lifestyle, Sex and Spay/Neuter Status According to Guardian Report

**DOI:** 10.3390/ani13121963

**Published:** 2023-06-12

**Authors:** Florencia Barrios, Gonzalo Suárez, Monique A. R. Udell, Juan Pablo Damián

**Affiliations:** 1Departamento de Clínicas y Hospital Veterinario, Facultad de Veterinaria, Universidad de la República, Montevideo 13000, Uruguay; florbarriosfernandez@gmail.com (F.B.); suarezveirano@gmail.com (G.S.); 2Department of Animal and Rangeland Sciences, Oregon State University, Corvallis, OR 97331, USA; monique.udell@oregonstate.edu; 3Departamento de Biociencias Veterinarias, Facultad de Veterinaria, Universidad de la República, Montevideo 13000, Uruguay

**Keywords:** demography, domestic felines, *Felis catus*, Latin America, survey

## Abstract

**Simple Summary:**

Despite the fact that the domestic cat is one of the most popular pets in the world, there is little information (at the country level) in Latin America on its demographic characteristics. Therefore, the objective of this study was to characterize the domestic feline population of Uruguay in relation to breeds, coat colors, hair length (long vs. short), lifestyle (indoor vs. outdoor), age, sex, and spay/neuter status represented across the country according to a survey of guardians. The population of cats with guardians in Uruguay is characterized by the following data: higher frequency of female cats (53%), most of the cats were between two and six years old (49%), most of them were neutered (84%, mainly those older than one year of age), most of them have outdoor access (87%), a very low percentage (6%) are purebred (Siamese being the most frequent: 86%), and within the non-pure breeds, short hair cats were the most frequent (79%). This is the first study in Latin America to describe some key demographic aspects such as cat breeds, coat color, hair length, lifestyles, and frequency by age, sex, and spay/neuter status (spayed/neutered or not) at the country level.

**Abstract:**

The objective of this study was to characterize the domestic cat population of Uruguay in relation to breed, coat color, hair length, lifestyle (indoor vs. outdoor), age, sex, and spay/neuter status according to a survey completed by their owners or guardians. An online survey, distributed to residents of Uruguay, was completed in full by 2561 cat guardians. Descriptive statistics and Chi-squared tests were performed. The population of cats with guardians in Uruguay is characterized by the following data: higher frequency of female cats (53%), most of the cats were between two and six years old (49%), most of them were neutered (84%, mainly those older than one year of age), most of them have outdoor access (87%), a very low percentage (6%) are purebred (Siamese being the most frequent: 86%), and within the non-pure breeds, short hair cats were the most frequent (79%). This study, in addition to expanding the information on the characteristics of cats with guardians from other countries and continents, is the first study in Latin America to describe some key demographic aspects such as cat breeds, coat color, hair length, lifestyles, and frequency by age, sex, and spay/neuter status (spayed/neutered or not) at the country level.

## 1. Introduction

Today, the domestic cat (*Felis catus*) is one of the most popular pets in the world [1,2]. In addition to their growth in popularity, the role of the domestic cat in human life has changed in recent years, going from being considered an almost independent animal to becoming part of the family. A study conducted in the Netherlands found that of 1859 guardians, 52% considered their cat to be a member of the family, 27% considered it a child and 7% considered it their best friend [3]. In that sense, it is evident that many guardians form a strong and stable emotional bond with their cats [3,4], and that this bond has potential advantages for both species [5,6,7,8]. 

Knowing and characterizing animal population data related to demography and lifestyles, such as whether they live indoors or outdoors, their breed, age, coat length, coat color, sex distribution, and spay/neuter status, are key elements for animal health and welfare purposes [9,10,11]. For example, the percentage of neutered cats, as well as those that live outdoors, are necessary data to evaluate population control, their environmental impact, and the dynamics of zoonoses and other diseases in whose cycle they are involved. Cats and dogs that live outdoors, with or without guardians, experience greater health risks than their indoor-living counterparts, and the overpopulation of free-roaming animals can result in other challenges, including traffic accidents, bite injuries, predation, and competition with wildlife [12,13,14,15,16,17,18]. The decision to allow cats access to the outdoors has not been found to be associated with their age, health status, or onychectomy status. However, cats with outdoor access are twice as likely to have signs related to early degenerative joint disease than those with only indoor access [19]. Additionally, cats allowed outdoor access are more likely to be bitten by other cats [20], and are almost three times more likely to become infected with parasites than those indoors [21]. Consequently, this status can have important implications for cat welfare. The behavior of cats that live outdoors compared to those that live indoors can also differ. For example, outdoor cats have been found to cover more distance and consume more food than indoor cats, but show fewer rhythmic behaviors [22]. In addition, especially in countries such as Colombia, with a great diversity of wild species, it is estimated that domestic cats kill between three and twelve million birds annually in urban and suburban areas, so cat access to the outdoors may also negatively influence the ecosystem [23]. However, outdoor access also has benefits for animal welfare, by allowing cats to display natural behaviors for the species, such as hunting, exploring, and climbing [24]. The number of outdoor cats varies considerably according to their geographic location, with cats more likely to be outdoors in countries within Europe and Oceania than in North American countries, such as the US and Canada [10,24,25,26], so it is especially important to conduct research evaluating these trends across different countries and populations. In South America, there are only three studies that have evaluated the frequency of indoor and outdoor cats, and these were all conducted in the same country (Brazil). However, even within one country, data collected in different states produced contradictory results. A study conducted by Canatto et al. [27] in the state of São Paulo found that cats were more frequently housed indoors than outdoors, while a study conducted by Felipetto et al. [28] in the State of Rio Grande do Sul and a study by Trapp et al. [29] conducted in the State of Paraná (city of Jaguapitã), found that more cats lived outdoors than indoors in these regions. However, to date, there are no studies that have been carried out at the level of a whole country in Latin America.

The demographic representation of breeds, coat color, and hair length of cats may also vary according to the continents and countries represented in the literature. While in Europe and North America, the predominant breeds are the British Shorthair, Siamese/Oriental, Persian, Maine Coon, and Bengal [30,31,32], in Oceania (Australia and New Zealand), the predominant breed is the Burmese [10,33], and in Asia (Pakistan), the most dominant breeds were Persian, Bombay, Turkish van cats and Maine Coon [34]. Although descriptive data on cat breeds, hair length, and coat color have been evaluated in several countries in North America, Europe, Oceania, and Asia, data from Latin American countries are lacking. In addition, the risk factor for certain diseases varies according to cat breed. The Burmese breed is a risk factor for diabetes, lipid aqueous, and pancreatitis [35], while cats of the Maine Coon breed are prone to hip dysplasia and osteoarthritis [36,37]. 

Despite the importance of what the cat represents as a pet for humans, there is a lack of general information at the country level in Latin America on basic elements, such as the most frequent breeds, hair length, varieties of coat colors, and lifestyle, among other features. Although there are few studies on cat demography in South America, these were carried out at the state or city level and not at the country level (such as in the State of São Paulo or Rio Grande do Sul, Brazil: [27,28,38,39]; in Santiago city, Chile: [40]; in Bucaramanga city, Colombia: [41]). These studies also evaluated targeted elements within cat populations, such as the frequency of males and females, whether they were neutered or spayed, or the proportion of animals in relation to humans. However, according to our knowledge, other important characteristics of cat populations, such as the presence and representation of different breeds, hair length, and coat colors, have not been studied in Latin America. Therefore, the objective of this study was to broadly characterize the domestic feline population of Uruguay in relation to breeds, coat colors, hair length (long vs. short), lifestyle (indoor vs. outdoor), age, sex, and spay/neuter status represented across the country according to a survey of owners or guardians.

## 2. Materials and Methods

The present study consisted of the implementation and subsequent analysis of a cross-sectional, online survey created in Google Forms™ [42]. This platform was chosen due to its free access and its proven effectiveness in conducting research surveys [43]. Both the dissemination and completion of the survey were completely online. Considering that in Uruguay, 88% of the population has Internet access at home [44], it was anticipated that this method would allow a significant range of cat guardians to access it. To complete the form, cat guardians had to be at least 18 years old, be Uruguayan residents, and be the guardian of at least one cat. The survey, in its final version, began receiving responses in March 2020 and was closed in July 2020. According to what was reported by Ruiz et al. [45]: “This study was designed and conducted in adherence to the indications of the Declaration of Helsinki. Privacy and data confidentiality were maintained throughout the process. A specific ethical agreement is not needed in Uruguay for the type of survey employed”. The survey complies with the data security requirements framed in Law No. 18331 (Uruguay). The guardian’s full name and email were collected in each survey. These data were used to contact them in case of doubts about survey data or to subsequently request authorization for the use of the photos attached to the survey in case the research team considered them useful. The data was encrypted and stored to continue working on subsequent publications related to this survey, according to their respective consents.

### 2.1. Survey Design

The questions used in this study were divided into 5 main blocks: (1) Guardian demographic information, (2) cat’s basic information: age, sex, and spay/neuter status (neutered/spayed or unneutered/unspayed), (3) physical characteristics: breed and coat characteristics, (4) home, lifestyle, environment, and coexistence data, and (5, optional) suggestions and cat photo upload. Most of the questions were multiple choice, multiple answer, or binary response type (Yes/No), although some had the options “Do not know/Does not answer” and “Others” (the latter allowing free written response). The survey was distributed only in Spanish. The terms were presented with scientific language, but clarifications in colloquial language were attached to terms that may have been poorly understood by the general public. Repeated or incomplete surveys were not considered for this study. The specific questions used for this article are presented in Table 1. For the present work, the cats were classified as indoor if their guardians reported that they do not have any access to the exterior of the property, at any time. Cats that access the exterior of the property were classified as outdoor. In the latter case, no distinctions were made regarding the degree of access to the exterior of the property (more or less supervised).

### 2.2. Target Audience (Population of Interest)

Sample size calculations were made based on the estimated percentage of Uruguayan guardians (68%) who had at least one cat [46] and the total number of inhabitants in Uruguay (3,252,091 people according to the Official Census) [47]. We considered a confidence interval of 95% and a margin of error of 5% and obtained a sample result of (at least) 385 cat guardians in order to obtain representative information for this population. The number of completed surveys received from cat guardians in this study was 2572, representing a margin of error of 0.27%. Of the 2572 surveys, 2561 of them met the conditions to be included in this research.

### 2.3. Analysis and Review of Surveys

First, each survey was reviewed individually, corroborating its viability (complete responses) and compliance with the guardian’s participation requirements. Then, each survey was analyzed and accounted for according to the previously established variables, also classifying the “Other” responses (those that allowed a written response). All the surveys were reviewed individually, and the guardian was contacted to corroborate any doubtful data. 

### 2.4. Statistical Analysis

A descriptive analysis of sex, age, breed, coat colors, hair length, lifestyle (outdoor vs. indoor), and spay/neuter status was carried out to determine the percentage of cats with each trait. Statistical differences in the frequency of cats according to sex, age, spay/neuter status, breeds, coat colors and hair length, lifestyle, and associations were calculated using a Chi-square test (with a significance of 95%). All analyses were performed in R (Version 4.1.2) [48,49].

## 3. Results

### 3.1. Sex, Age, and Spay/Neuter Status 

From a total of 2551 cats, the percentage of female cats was greater than male cats (females: 53% vs. males: 47%; Chi-Square 10.67, df1, *p* < 0.0001). Only 10 guardians reported not knowing the sex of their cats. 

Regarding the age of the cats, 49% of cats were between two to six years old, 28.8% were under one year of age, and the remaining 20.5% were over seven years (Chi-Square 1796, df9, *p* < 0.01). Only 1% of guardians reported not knowing the age of their cat. The complete distribution by age and sex of cats is shown in Figure 1 (The ‘unknown’ age responses were not included in Figure 1). The number of cats by age was associated with the sex (Chi-Square 20.23, df169, *p* = 30.04), with a greater number of females, compared to males, represented in the following age groups: 7 to 10 years (Chi-Square 6.776, df1, *p* = 0.009); 11 to 15 years (Chi-Square 6.2, df1, *p* = 0.01) and more than 15 years (Chi-Square 7.811, df1, *p* = 0.005).

Regarding the spay/neuter status of the cats, most of the guardians (84%) reported that their cats were spayed or neutered (Chi-Square 1149, df1, *p* < 0.01). The distribution of spay/neuter status in relation to age and sex is shown in Figure 1. Between three months and six years of age, there was no difference in the proportions of neutered and unneutered cats for both sexes, with the proportion of sterilized females being higher after seven years of age. Only 11 guardians stated that they were unaware of their cat’s spay/neuter status. Within the first year of life, regardless of sex, most cats were spayed or neutered (~90%) (Chi-Square 1182 (df9), *p* < 0.01). 

### 3.2. Lifestyle

A very low percentage of guardians (13%) indicated that their cats live indoors without outdoor access, in contrast to the high percentage of guardians who reported that their cat has access to the outdoors (87%) (Chi-Square 1369 (df1), *p* < 0.01)) (Figure 2) (‘Unknown’ age responses were not included in Figure 2). Outdoor or indoor cat classification was not significantly associated with breed, sex, or neuter status (*p* > 0.05). Figure 3 shows the distribution of guardian-reported breed, sex, and neuter status linked to the cat’s indoor/outdoor access.

Figure 3 provides a multilevel sankey diagram with four dimensions of the cats’ lifestyle (Environment, Breed, Sex, and Spayed), considering different orders of possible connection between dimensions. The first panel of Figure 3 illustrates the distribution of cat breed status, sex, and spay/neuter status for each lifestyle (indoor vs. outdoor cats). Subsequent panels further divide subgroups according to the third and fourth remaining dimensions (spay/neuter status, sex, and breed). In the first panel, it was observed that there was a stronger connection between lifestyle and the cats of non-breed than breed and then a proportionally similar distribution between sex and spay/neuter status. Similarly, the center panel shows similar connections by sex and spay/neuter status. Finally, the panel on the right shows a strong relationship to spay/neuter status, with a similar connection between sex and breeds. In summary, there were dimensions with greater weights when considering connections with indoor vs. outdoor lifestyle, such as non-breed classification and Spay/Neuter status.

### 3.3. Breeds and Hair Length

Regarding breeds, a high proportion of guardians reported that their cats were non-purebred (Non-purebred: *n* = 2415, 94% vs. Purebred: *n* = 146, 6%; Chi-Square 2010 (df1), *p* < 0.01). Figure 4 shows representative images of coat color patterns of non-pure breed cats and a Siamese as an example of a pure breed cat. 

As for the presence of cat breeds in Uruguay, the most common breed is the Siamese (86%), with the remaining 14% being distributed among the Persian (8.9%), Ragdoll (1.4%), Maine Coon (1.4%), Himalayan (0.7%), Bombay (0.7%), and Bengal (0.7%) breeds (Chi-Square = 623, df6, *p* < 0.01). There was no significant association between breeds and sex (Chi-Square = 7.56, df6, *p* = 0.27). 

Regarding the non-purebred cats and their hair length, we identified a higher percentage of cats with short hair (79%) compared to long hair (21%) (Chi-Square 831, df1, *p* < 0.01). An example of non-purebred cats with long and short hair is shown in Figure 5. There was no significant association between coat hair and sex (Chi-Square = 0.54. df1, *p* = 0.46). 

## 4. Discussion

### 4.1. Sex, Age, and Spay/Neuter Status 

This study provides the first estimates of cat sex frequencies in Uruguay, and the first carried out at the level of the entire country in Latin America. The fact that in Uruguay, the female sex is more frequent than the male sex in cats coincides with some reports from other countries (Australia: [10]; Italy: [11]; Brazil; [27,28] and Ethiopia: [50]), while other studies have found no differences in the frequency of male and female cats (Brazil: [44]; EE.UU.: [51]; Australia and New Zealand: [33]; Great Britain [32]) or that males were more frequent than females (Colombia: [41]; Chile: [40]; EE.UU: [52]; Netherlands [3]). This suggests that the frequency of cats by sex can vary according to the different geographical regions, countries, and continents surveyed. In South America, it has been observed that the frequency of each sex varies according to the country and even within the same country, highlighting the need to generate data at the country level on the demographic aspects of cats. 

Differences between the sexes in individuals older than seven years could be linked to longevity. For example, O’Neill et al. [31] in England, reported that female cats had a higher median lifespan than male cats (F: 15.0 years vs. M: 13.0 years); this is also the case of Kent et al. [53] in the United States who found the same difference. Although the results reported by O’Neill et al. [31] are in line with our study, other studies have found no effects of sex on the longevity of cats [54].

There is little information on the percentage of neutered cats in Latin America, especially in South America. The frequency of neutered or spayed cats in Uruguay is greater than reported in some Brazilian cities [28,39]. In addition, the percentage of neutered or spayed cats in Uruguay is greater than in some countries, such as Italy [11], Ireland [55], United Kingdom [32] or Ethiopia [50], and less than in other countries, such as the USA [52], Australia and New Zealand [33]. According to some authors, between a 75% and 90% sterilization rate should be high enough to control a pet overpopulation problem, based on a model developed by Nassar and Mosier (1980 and 1982) [56] and Andersen et al. [57]. Regarding pet neutering policies in Uruguay, in February 2023, the Uruguayan government decreed the creation of the National Reproductive Control Program (PNCR) and ordered the mandatory sterilization of all dogs and cats of both sexes and throughout the territory, a task that will be carried out by the National Institute of Animal Welfare (Instituto Nacional de Bienestar Animal—INBA). Given the novelty of this decree, its implementation is not yet clear, and it is still not a reality, so to date, the decision to neuter/spay their cat is exclusive to the guardian. Therefore, the results regarding the percentage of neutered/spayed cats in Uruguay reinforce the concept that their guardians are aware of their responsibility and role in the management and control of the population of these animals [58].

Within the first year of life, regardless of sex, most cats were spayed or neutered. In this sense, cat guardians in Uruguay prefer to neuter/spay their cats rather than leave them intact, and most of them do so before the cat turns one year old. Although most owners neuter/spay their cats around one year, it is important to note that cats can be fertile at 5–6 months of age.

### 4.2. Lifestyle

The indoor/outdoor access of cats varies across continents. In countries of Europe, Oceania, and Africa, most cats have access to the outdoors [10,11,24,33,50,55], while in North America (especially the USA and Canada), a low percentage of cats have access to the outdoors [20,24,55]. Based on these data, in Uruguay, the cat’s lifestyle follows a more similar pattern than that of countries in Europe, Oceania, and Africa. However, studies carried out in Brazil (but in different regions and cities) found differences in the percentage of outdoor cats, which ranged from 42.5% [27], 66.8% [28], to 80% [29]. These results highlight that there are not only differences between continents and countries, but also within the same country, where percentages of indoor/outdoor cats can vary from one region to another. In any case, two of the three studies carried out in Brazil showed a preference for allowing cats outdoors, which is consistent with the data from Uruguay. The fact that the majority of cats in Uruguay have free access to the outdoors can represent some health and welfare disadvantages for them, such as the increased risk of diseases, parasites, accidents, and traffic deaths, as well as a threat to wildlife due to predatory behaviors (see review: [24]). On the other hand, as also mentioned by Tan et al. [24], outdoor access may provide welfare benefits to cats, such as the possibility of displaying behaviors typical of the species, such as exploring, hunting, and climbing.

### 4.3. Breeds and Hair Length

Although the percentage of purebred and non-purebred cats varies between countries, in all published reports, the percentage of non-purebred cats has varied between approximately 70% and 95%, which is much higher than purebred cats. Within this range, Uruguay is among the countries with the highest percentage of non-purebred cats, such as Italy (95.6%) [11], Colombia (94.1%) [41], and USA (90.9%) [52], and has a greater non-purebred percentage when compared with countries such as Australia and New Zealand (72.5−76.5%) [10,33], and Brazil (81.2%) [29]. In other regions, such as the United Kingdom, the results have varied from 77.8% [30], to 89.6% [32], or 92.9% [31].

According to our knowledge, this study is also the first to report data on the percentage of cat breeds in Latin America. In Uruguay, the most common breed is the Siamese, with the remaining being distributed among the Persian, Ragdoll, Maine Coon, Himalayan, Bombay, and Bengal breeds. The profile of cat breeds in Uruguay is more similar to those reported in Europe and North America [30,31,32] than in other countries and continents such as Oceania (Australia and New Zealand [10,33] and Asia (Pakistan [34]). What does stand out clearly is the high percentage of Siamese (Figure 4J) as the main breed. Compared to other countries with survey data on the presence of different breeds of cats, Uruguay is the country that most frequently has Siamese cats. On the other hand, the second most frequent breed in Uruguay is the Persian, which is also frequent in Europe, North America [30,31,32], and Asia [34], but not in Oceania. Another interesting difference is that in Uruguay, the Burmese breed was not reported by guardians, while this is the most frequent breed reported in Oceania (Australia and New Zealand) [10,33]. This information reaffirms that there are differences in the presence and number of purebred cats between countries and continents, and Uruguay within Latin America presents a frequency and profile of breeds different from others. Therefore, this study highlights the need to know the frequency of cat breeds in other countries, information that may be influenced by cultural, social, political, and demographic differences depending on the region.

Only a few prior studies have surveyed the number of cats with long or short hair in other countries and regions of the world. In Oceania, Toribio et al. [10] reported that shorthair cats were the most common (89.8%) in Australia, while Johnston et al. [33] also reported that short hair cats were more common (55%) in Australia and New Zealand. Trapp et al. [29] reported that shorthair cats were also the most frequent (92.7%) in a region of Brazil. The frequency of non-purebred and short-haired cats in Uruguay matches the percentages of the works mentioned above, where they represent a vast majority in contrast to those long-haired cats.

In relation to this study, it is important to be cautious with the results concerning some considerations: (a) the survey was carried out by a subpopulation of people with access to mobile telephony (wireless phone/internet access), leaving out those who did not have access to it, (b) no comparisons were made between urban and rural cats, and (c) it is important to highlight that the survey was carried out during the COVID-19 pandemic, although it would not have significant effects on the characteristics of the cat population described in this study, we cannot rule out possible social influences caused by the aforementioned Pandemic.

## 5. Conclusions

This study represents the first demographic characterization of domestic cats with guardians in Uruguay. The surveyed population reported on cats characterized by the following data: higher frequency of female cats, most of the cats were between 2 and 6 years old, most of them were neutered (84%, representing most cats older than one year of age), most of them have outdoor access (87%), a very low percentage (6%) are purebred (Siamese being the most frequent: 86%), and within the non-purebred, short hair cats were the most frequent (79%). This study, in addition to compiling information on the characteristics of cats with guardians from other countries and continents, is the first study in Latin America to describe at the country level some key demographic aspects such as cat breeds, coat color, hair length, lifestyles and frequency by age, sex, and spay/neuter status. Such information can help improve understanding of population similarities and differences and is important for the development of targeted cat management practices used to improve the health and welfare of animals across different regions. 

## Figures and Tables

**Figure 1 animals-13-01963-f001:**
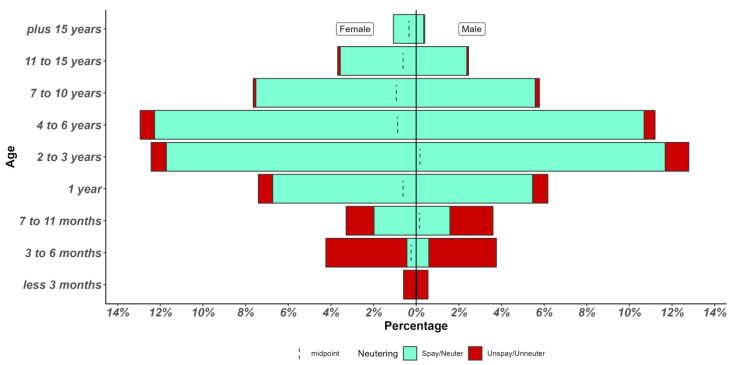
Age, sex, and spay/neuter status distribution of the cats that the guardians reported in the survey.

**Figure 2 animals-13-01963-f002:**
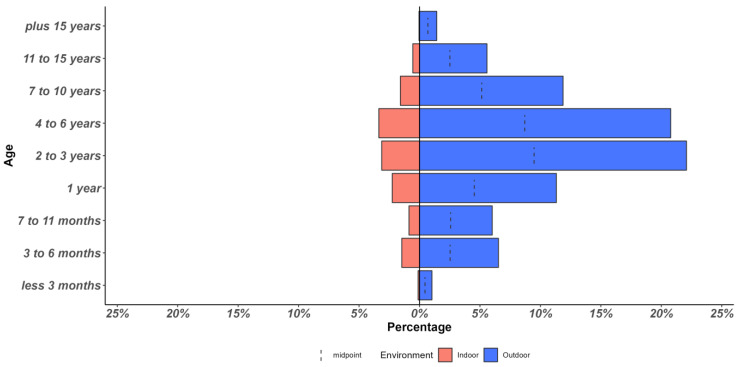
Age and lifestyle (indoor vs. outdoor) distribution of the cats reported by the guardians in the survey.

**Figure 3 animals-13-01963-f003:**
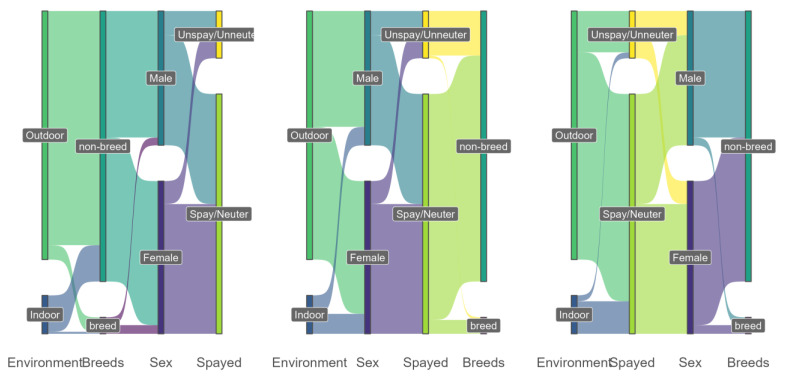
Breeds, sex, and neuter/spay status connection of the cats, considering indoor/outdoor access reported by their guardians in the survey.

**Figure 4 animals-13-01963-f004:**
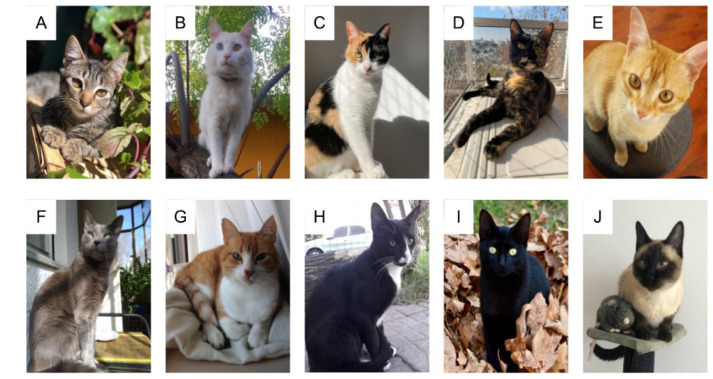
Description of coat color patterns included in the survey: Tabby (**A**), Completely white (**B**), Calico (**C**), Carey (**D**), Orange/Yellow (**E**), Completely grey (**F**), White-spotted (**G**), Black and White (**H**), Completely black (**I**) and Color according to breed (as an example a Siamese cat is shown (**J**)). The photos shown are with copyright permission.

**Figure 5 animals-13-01963-f005:**
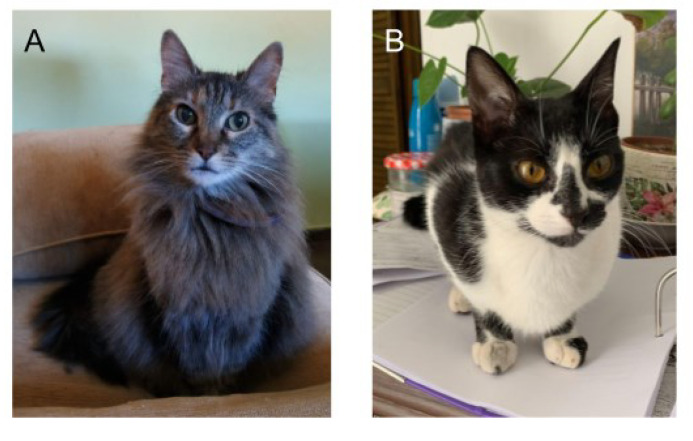
Description of non-purebred coat length included in the survey, long-haired non-purebred breed (**A**), and short-haired non-purebred (**B**). The photos shown are with copyright permission.

**Table 1 animals-13-01963-t001:** The survey questions included in this article are described below. These include questions regarding breeds, coat lengths and colors, lifestyle, sex, and spay/neuter status according to the guardians. The type of answer presented is clarified under each question. Answers could be of the type “Numeric field”, “Multiple choice”, “Multiple answer” or “Open question”.

Sex and spay/neuter status (Multiple choice)
Age (Numeric field)
If the cat was spayed/neutered, at what age was it performed? (Multiple choice)
Breed and coat length (Multiple choice)
Hair Color (Multiple choice). This question had a representative photo of each option presented (as seen below)
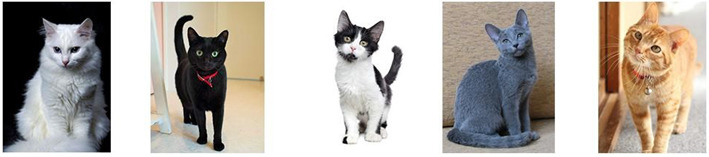
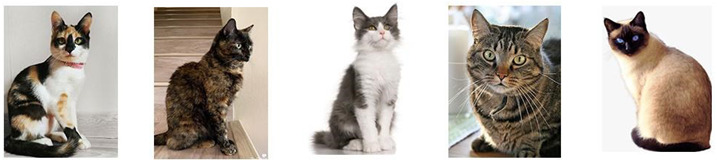

## Data Availability

The data used for this study are available from the corresponding author upon reasonable request.

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
