# Peer review of "Characterization of the Domestic Cat Population of Uruguay: Breeds, Coat Colors, Hair Length, Lifestyle, Sex and Spay/Neuter Status According to Guardian Report"

_animals, 2023, doi:10.3390/ani13121963_

Round 1

Reviewer 1 Report

Comments on the manuscript “Characterization of the domestic cat population of Uruguay: Breeds, coat colors, hair length, lifestyle, sex and spay/neuter status according to guardian report” submitted to the Animals

General comments

I thank the editor for the opportunity to comment on the manuscript entitled “Characterization of the domestic cat population of Uruguay: Breeds, coat colors, hair length, lifestyle, sex and spay/neuter status according to guardian report”.

The article describes the phenotypic and reproductive profile of domestic cats in Uruguay and some ways of raising them. The writing is very understandable, showing important data for areas of knowledge such as veterinary medicine, ecology, anthrozoology, economics, and others.

The article becomes more relevant because the Uruguayan health authorities enacted a new pet neutering policy in 2023 (The National Reproductive Control Program). Therefore, all dogs and cats of both sexes must be sterilized.

Despite all the article's positive aspects, I am worried about some gaps. First of all, it is surprising that the authors did not mention an explicit statement that the research received a license from research ethics committees because Uruguay is a signatory of international agreements for the protection and well-being of animals and humans. Second, it is regrettable that guardian data was not collected; or if they were collected, that they are not analyzed. The analysis of the cats' data with the guardians' data would have a more profound effect on the understanding of the human-animal relationship, in the South American social profile represented by the Uruguayan population. Finally, my few comments and suggestions I present below:

Line 13: Does not need “today” word.

Line 39: Why not insert the word Felis catus? This is not a study strictly involving Latin America, but there are comparisons across continents. I suggest organizing the keywords in alphabetical order.

Line 58: What kind of noise at night can be bad for cats?

Line 170: Explain: How was the population of interest chosen? How was the questionnaire distributed?

Line 195: Start by showing how many cats were sampled.

If the sentence begins by stating that the result is in frequency, why present the proportion and percentage? Note that in Figures 1 and 2, the results are presented in percentages. I find it confusing to write this way: females: 1358/2551; 53% vs. males: 1193/2551; 47%; Chi-Square 10.67, df1, p < 0.0001. It seems more understandable to write like this: females: 53% vs. males: 47%; Chi-Square 10.67, df1, p < 0.0001.

Lines 206 and 225; Figures 1 and 2: The vertical axis has the age classes of cats in sequential order. The last one, higher up (unknown) is a category that deviates from the logic of the time interval (age classes) of the axis. Therefore, I recommend that the "unknown" category be mentioned in the text, without necessarily being inserted in the figure.

Line 327, Figure 3: Figure 3 is ingenious but counterintuitive. It will be difficult for readers of the article to understand. I suggest modifying it to a more understandable format.

Line 242: I think “purebred” is missing (Non-purebred: n = 2415, 94% vs. ?n = 146, 6%; Chi-Square 2010 (df1), p < 0.01). Delete the “df” parentheses.

Lines 248 and 253: The Chi-square is not a statistical association test. Therefore, the word must be changed.

Line 356: I suggest that some expressions of caution about the results be introduced in the text. This caution has to do with the type of sampling, that is, how the respondents were selected. The subpopulation that does not have access to mobile phone services or that does not have the ability to respond to the questionnaire was left out of the sample. Another problem is the lack of comparisons between urban and rural cats.

Another aspect of the research that the authors seem to give less importance to is the period in which the survey was carried out, which coincides with the peak of the COVID-19 pandemic. What impact could social changes during the COVID-19 pandemic have on research? It is important to comment on this.

Author Response

(R) Reviewer 1:

General comments

I thank the editor for the opportunity to comment on the manuscript entitled “Characterization of the domestic cat population of Uruguay: Breeds, coat colors, hair length, lifestyle, sex and spay/neuter status according to guardian report”.

The article describes the phenotypic and reproductive profile of domestic cats in Uruguay and some ways of raising them. The writing is very understandable, showing important data for areas of knowledge such as veterinary medicine, ecology, anthrozoology, economics, and others.

The article becomes more relevant because the Uruguayan health authorities enacted a new pet neutering policy in 2023 (The National Reproductive Control Program). Therefore, all dogs and cats of both sexes must be sterilized.

  • Authors:

The authors thank the reviewer for the positive comments regarding the manuscript and for highlighting its important aspects.

R: Despite all the article's positive aspects, I am worried about some gaps. First of all, it is surprising that the authors did not mention an explicit statement that the research received a license from research ethics committees because Uruguay is a signatory of international agreements for the protection and well-being of animals and humans.

A: We appreciate the reviewer's comment and concern on this issue. As previously reported to the editor, we made the following clarifications:

We submitted an inquiry to the Ethics Committee for the Use of Humans in Uruguay about the work carried out by our team and it was determined that our work did not require evaluation by an Ethics Committee for the Use of Humans (Exempt Status). The justification of the Committees is based on the fact that: 1) in our study no biological sample was collected from any person, 2) that it is only an anonymous survey (as can be seen in the manuscript submitted to the journal), 3) that the survey was not considered human subjects research as it did not collect data about the person, but on the cats they have as pets, 4) that the survey does not collect any information "sensitive" by the people, 5) in the survey it was made clear to the people that the information is for a scientific study.

Lastly, some photos of cats appear in the manuscript, and the photos of all these cats were provided by the owners or keepers with their respective consent, giving due permission to use them in this scientific article. This document was attached in the process of submitting the manuscript to the journal.”

In addition, we included the following sentences in the manuscript: According to what was reported by Ruiz et al. [45]: “This study was designed and conducted in adherence to the indications of the Declaration of Helsinki. Privacy and data confidentiality were maintained throughout the process. A specific ethical agreement is not needed in Uruguay for the type of survey employed." The survey complies with the data security requirements framed in Law No. 18331 (Uruguay). L121-126.

Based on that previous explanation, the Editor answered us in the affirmative as shown below: "We are glad to share with you that the Academic Editor considered the ethical issue is fine for this manuscript based on your explanation.

Reference:

Ruiz P, Semblat F, Pautassi RM. Change in Psychoactive Substance Consumption in Relation to Psychological Distress During the COVID-19 Pandemic in Uruguay. Sultan Qaboos Univ Med J. 2022;22(2):198-205. doi:10.18295/squmj.5.2021.106.

R: Second, it is regrettable that guardian data was not collected; or if they were collected, that they are not analyzed. The analysis of the cats' data with the guardians' data would have a more profound effect on the understanding of the human-animal relationship, in the South American social profile represented by the Uruguayan population.

A: We understand the reviewer's approach, and agree that further analysis of guardian data would be interesting. We plan to do further analyses considering such additional factors in the future. For this manuscript questions were chosen based on the current scope of objectives for which results are presented. Other questions that are linked to other topics of interest are being analyzed to be presented in a future paper.

R: Finally, my few comments and suggestions I present below:

Line 13: Does not need “today” word.

A: The word "today" was removed, thanks for the suggestion.

R: Line 39: Why not insert the word Felis catus? This is not a study strictly involving Latin America, but there are comparisons across continents. I suggest organizing the keywords in alphabetical order.

A: Thanks for the suggestions. We include the word "Felis catus" and organize the keywords alphabetically. L38.

R: Line 58: What kind of noise at night can be bad for cats?

A: We prefer to eliminate those words “noise at night” from the sentence that do not contribute to it to a great extent.

R: Line 170: Explain: How was the population of interest chosen? How was the questionnaire distributed?

A: The questionnaire was distributed through official communication media of the University of the Republic (social networks, institutional mail, billboards, student unions, among others) and through personal social networks of the members of this research work (Facebook, Instagram and WhatsApp). At the same time, another dissemination was achieved through the mass media (television) through interviews explaining the purposes of the investigation and the inclusion criteria of the guardians. The type of sampling was Snowball Sampling, since whoever completed it was encouraged to send it to contacts who might be interested in completing the survey as well.    

R: Line 195: Start by showing how many cats were sampled.

A: This change has been made as suggested. L186.

R: If the sentence begins by stating that the result is in frequency, why present the proportion and percentage? Note that in Figures 1 and 2, the results are presented in percentages. I find it confusing to write this way: females: 1358/2551; 53% vs. males: 1193/2551; 47%; Chi-Square 10.67, df1, p < 0.0001. It seems more understandable to write like this: females: 53% vs. males: 47%; Chi-Square 10.67, df1, p < 0.0001.

A: Since this manuscript consists of the description and characterization of the cat population, we wanted present them as a proportion and percentage, in order to give a complete view of such data and to for consistency with the analysis used (Chi-Square which relied on frequency and not percentage). We thank the reviewer for giving us the suggestion to write the sentence based on percentage alone. We have now edited the sentence as follows: “From a total of 2551 cats, the percentage of female cats was greater than male cats (females: 53% vs. males: 47%; Chi-Square 10.67, df1, p < 0.0001).” L186-188.

R: Lines 206 and 225; Figures 1 and 2: The vertical axis has the age classes of cats in sequential order. The last one, higher up (unknown) is a category that deviates from the logic of the time interval (age classes) of the axis. Therefore, I recommend that the "unknown" category be mentioned in the text, without necessarily being inserted in the figure.

A: Thanks for the suggestion, the change was made.

R: Line 327, Figure 3: Figure 3 is ingenious but counterintuitive. It will be difficult for readers of the article to understand. I suggest modifying it to a more understandable format.

A:  We appreciate the reviewer's comment. We expand the explanation of Figure 3 in the text. We believe that a Sankey diagram is a good visualization technique to show the relationship between dimensions (e.g., Environment, races, sex, and sterilized) rather than the entire relationship between elements across all entities. Their links are represented by arcs that have a width proportional to the importance of the specific dimensions.

     We rewrite the explanation of this figure as follows: “Figure 3 provides a multilevel sankey diagram with 4 dimensions of the cats lifestyle (Environment, Breed, Sex and Spayed), considering different orders of possible connec-tion between dimensions. The first panel of Figure 3 illustrates the distribution of cat breed status, sex and spay/neuter status for each lifestyle (indoor vs outdoor cats). Subsequent panels further divide subgroups according to the third and fourth re-maining dimensions (spay/neuter status, sex and breed). In the first panel, it was ob-served that there was a stronger connection between lifestyle and the cats of non-breed than breed and then a proportionally similar distribution between sex and spay/neuter status. Similarly, the center panel shows similar connections by sex and spay/neuter status. Finally, the panel on the right shows a strong relationship to spay/neuter status, with similar connection between sex and breeds. In summary, there were dimensions with greater weights when considering connections with indoor vs outdoor lifestyle, such as non-breed classification and Spay/Neuter status.” L221-233.

R: Line 242: I think “purebred” is missing (Non-purebred: n = 2415, 94% vs. n = 146, 6%; Chi-Square 2010 (df1), p < 0.01). Delete the “df” parentheses.

A: Yes, thanks for the observation. “Purebred” was incorporated L242.

R: Lines 248 and 253: The Chi-square is not a statistical association test. Therefore, the word must be changed.

A: The chi square test can be used to evaluate association between two categorical variables in some cases, see the following articles as examples: Franke et al. (2012), Scott et al. (2013), Rana and Singhal (2015), Kim (2017). In our case, we evaluated whether the categorical variables of cat breeds and sex were associated using this previously published approach.

     References:

     Franke, T. M., Ho, T., & Christie, C. A. (2012). The chi-square test: Often used and more often misinterpreted. American journal of evaluation, 33(3), 448-458.

     Kim, H. Y. (2017). Statistical notes for clinical researchers: Chi-squared test and Fisher's exact test. Restorative dentistry & endodontics, 42(2), 152-155.

Scott, M., Flaherty, D., & Currall, J. (2013). Statistics: Dealing with categorical data. Journal of Small Animal Practice, 54(1), 3-8.

Rana, R., & Singhal, R. (2015). Chi-square test and its application in hypothesis testing. Journal of the Practice of Cardiovascular Sciences, 1(1), 69.

R: Line 356: I suggest that some expressions of caution about the results be introduced in the text. This caution has to do with the type of sampling, that is, how the respondents were selected. The subpopulation that does not have access to mobile phone services or that does not have the ability to respond to the questionnaire was left out of the sample. Another problem is the lack of comparisons between urban and rural cats.

Another aspect of the research that the authors seem to give less importance to is the period in which the survey was carried out, which coincides with the peak of the COVID-19 pandemic. What impact could social changes during the COVID-19 pandemic have on research? It is important to comment on this.

A: To address the reviewer's comments, we now include the following paragraph in the discussion: “In relation to this study, it is important to be cautious with the results in relation to some considerations: a) the survey was carried out by a subpopulation of people with access to mobile telephony (wireless phone/internet access), leaving out those who did not have access to it, b) no comparisons were made between urban and rural cats, and c) it is important to highlight that the survey was carried out during the COVID-19 pandemic, although it would not have significant effects on the characteristics of the cat population described in this study, we cannot rule out possible social influences caused by the aforementioned Pandemic.” L360-367.

Reviewer 2 Report

This is a very interesting descriptive study about domestic cats in Uruguay. I have some minor comments concerning the study and article, which are mentioned hereafter. But first, I would like to express my concerns about the privacy of participants and safety of the datafile, as the survey was performed with Google Forms and participant personal data was processed [which I infer from (at line 184) 'the guardian was contacted to corroborate any doubtful data'].

You do not disclose by what type of manner participants are contacted (e-mail address, telephone number) and if names or other personal data was collected. You only mention in the ‘informed consent statement’ (line 380) that participants were informed about the purpose (science) and results will be presented in an anonymous manner.

Information about ensuring safety of the datafile, storage of the datafile, which personal data was collected and if personal data was deleted before storage, should be added disclosed in the article. At most universities an ethical committee makes sure that these issues are covered. However, as I miss an ethical committee approval statement in the article, it is of importance to disclose that participants data was handled while taken care of privacy and security.

Comments

Compliments to the first author for getting funding and publish an article based on an undergraduate thesis.

Line 59. Please rewrite this sentence: ‘The decision […..] is not à has not been found to be associated with […]. Otherwise it reads like an unreferenced claim.

Introduction. I miss the benefits of outdoor access for cats in terms of wellbeing and execution of natural feline behaviors like seeking, running, climbing, hunting/playing etc. The references only concern medical health issues. In the discussion the paper of Tan is mentioned. Please add this reference to the  introduction.

From line 94 differences in behavior and behavioral problems between breeds are mentioned. From this I expected that behavioral issues of breeds would be part of the survey. Consider omitting this paragraph from the introduction.

Results. Part 4.1

Add a sentence that although most owners neuter/spay their cats around 1 year, cats can be fertile at 5-6 months of age.

Figure 3. Consider changing Figure 3 as this it is not easily understandable. Rephrase ‘In Figure 3 we can see data…’ (line 228).

no comments

Author Response

(R) Reviewer 2:

Comments and Suggestions for Authors

This is a very interesting descriptive study about domestic cats in Uruguay. I have some minor comments concerning the study and article, which are mentioned hereafter. But first, I would like to express my concerns about the privacy of participants and safety of the datafile, as the survey was performed with Google Forms and participant personal data was processed [which I infer from (at line 184) 'the guardian was contacted to corroborate any doubtful data'].

You do not disclose by what type of manner participants are contacted (e-mail address, telephone number) and if names or other personal data was collected. You only mention in the ‘informed consent statement’ (line 380) that participants were informed about the purpose (science) and results will be presented in an anonymous manner.

Information about ensuring safety of the datafile, storage of the datafile, which personal data was collected and if personal data was deleted before storage, should be added disclosed in the article. At most universities an ethical committee makes sure that these issues are covered. However, as I miss an ethical committee approval statement in the article, it is of importance to disclose that participants data was handled while taken care of privacy and security.

(A) Authors:

We thank the reviewer for finding this study interesting.

We appreciate the reviewer's comment and concern on this issue.

As previously reported to the editor, we made the following clarifications:

We submitted an inquiry to the Ethics Committee for the Use of Humans in Uruguay about the work carried out by our team and it was determined that our work did not require evaluation by an Ethics Committee for the Use of Humans (Exempt Status). The justification of the Committees is based on the fact that: 1) in our study no biological sample was collected from any person, 2) that it is only an anonymous survey (as can be seen in the manuscript submitted to the journal), 3) that the survey was not considered human subjects research as it did not collect data about the person, but on the cats they have as pets, 4) that the survey does not collect any information "sensitive" by the people, 5) in the survey it was made clear to the people that the information is for a scientific study.

Lastly, some photos of cats appear in the manuscript, and the photos of all these cats were provided by the owners or keepers with their respective consent, giving due permission to use them in this scientific article. This document was attached in the process of submitting the manuscript to the journal.”

In addition, we included the following sentences in the manuscript: According to what was reported by Ruiz et al. [45]: “This study was designed and conducted in adherence to the indications of the Declaration of Helsinki. Privacy and data confidentiality were maintained throughout the process. A specific ethical agreement is not needed in Uruguay for the type of survey employed." The survey complies with the data security requirements framed in Law No. 18331 (Uruguay). L121-126.

Based on that previous explanation, the Editor answered us in the affirmative as shown below: "We are glad to share with you that the Academic Editor considered the ethical issue is fine for this manuscript based on your explanation.

Reference:

Ruiz P, Semblat F, Pautassi RM. Change in Psychoactive Substance Consumption in Relation to Psychological Distress During the COVID-19 Pandemic in Uruguay. Sultan Qaboos Univ Med J. 2022;22(2):198-205. doi:10.18295/squmj.5.2021.106.

R: Comments

Compliments to the first author for getting funding and publish an article based on an undergraduate thesis.

A: Both the first author and the co-tutors (co-authors) of this study feel very gratified by the work done and we thank the reviewer for the very nice words.

R: Line 59. Please rewrite this sentence: ‘The decision […..] is not à has not been found to be associated with […]. Otherwise it reads like an unreferenced claim.

A: The change has been made as suggested: “The decision to allow cats access to the outdoors has not been found to be associated with their age, health status, or onychectomy status.”, L58-59.

R: Introduction. I miss the benefits of outdoor access for cats in terms of wellbeing and execution of natural feline behaviors like seeking, running, climbing, hunting/playing etc. The references only concern medical health issues. In the discussion the paper of Tan is mentioned. Please add this reference to the introduction.

A: In accordance with the reviewer's comment, we now include the following statement in the introduction: “However, outdoor access also has benefits for animal welfare, by allowing cats to dis-play natural behaviors for the species, such as hunting, exploring and climbing [24].” L70-72.

R: From line 94 differences in behavior and behavioral problems between breeds are mentioned. From this I expected that behavioral issues of breeds would be part of the survey. Consider omitting this paragraph from the introduction.

A: Based on the reviewer's comment, we removed the following sentences from the introduction: “In addition, differences in behavior and behavior problems according to breed and/or hair color have been reported. Cat owners and veterinarians consider Persian cats less playful and active than Siameses [38-40]. Wilhelmy et al. [40] reported that Abyssinian cats have high scores for sociability and aggression, while Birman cats had higher scores for play, activity, vocalization, predatory behavior, and fear-related aggression. British Shorthair, Persian and Cornish Rex cats were least likely to be aggressive towards strangers, while Turkish Van cats were most likely to be aggressive towards other cats and Russian Blue cats had the highest probability for shyness towards strangers as opposed to Birman cats [41]. Lilac-coated cats had lower prey interest and higher play scores, while red-coated cats were more likely to exhibit fear-related aggression toward unfamiliar people, and tortoiseshell-coated (Carey) cats were associated with increased cat aggression scores and interest in prey [40]. Some research has found that cat owners are more likely to describe orange cats as friendly and tri-colored cats as intolerant [42]. Both behavior differences and guardian beliefs about breed/behavior differences, are factors that may differ across countries and regions and are important to animal welfare outcomes.”

R: Results. Part 4.1

Add a sentence that although most owners neuter/spay their cats around 1 year, cats can be fertile at 5-6 months of age.

A: Based on the reviewer's comment, we include the following sentence in the discussion: “. Although most owners neuter/spay their cats around 1 year, it is important to note that cats can be fertile at 5-6 months of age.” L304-305.

R: Figure 3. Consider changing Figure 3 as this it is not easily understandable. Rephrase ‘In Figure 3 we can see data…’ (line 228).

A: We appreciate the reviewer's comment. We expand the explanation of Figure 3 in the text. We believe that a Sankey diagram is a good visualization technique to show the relationship between dimensions (e.g., Environment, races, sex, and sterilized) rather than the entire relationship between elements across all entities. Their links are represented by arcs that have a width proportional to the importance of the specific dimensions. However, we have now provided more detail in the text to better help explain what Figure 3 depicts and have also reworded the noted sentence as suggested.

     We rewrite the explanation of this figure as follows: “Figure 3 provides a multilevel sankey diagram with 4 dimensions of the cats lifestyle (Environment, Breed, Sex and Spayed), considering different orders of possible connec-tion between dimensions. The first panel of Figure 3 illustrates the distribution of cat breed status, sex and spay/neuter status for each lifestyle (indoor vs outdoor cats). Subsequent panels further divide subgroups according to the third and fourth re-maining dimensions (spay/neuter status, sex and breed). In the first panel, it was ob-served that there was a stronger connection between lifestyle and the cats of non-breed than breed and then a proportionally similar distribution between sex and spay/neuter status. Similarly, the center panel shows similar connections by sex and spay/neuter status. Finally, the panel on the right shows a strong relationship to spay/neuter status, with similar connection between sex and breeds. In summary, there were dimensions with greater weights when considering connections with indoor vs outdoor lifestyle, such as non-breed classification and Spay/Neuter status.” L221-233.

Round 2

Reviewer 1 Report

Dear Editor

I appreciate the opportunity to review this new version of the manuscript. The authors modified the manuscript, accepting almost all of my suggestions. The lack of clarification on ethical procedures was answered. Other methodological aspects and presentation of results were answered and corrected, when necessary. Some caveats about study limitations were inserted in the text.

From my point of view, the manuscript has improved substantially and is well written to be accepted.

Sincerely, 

Author Response

The authors thank the reviewer again for the work done, and for the positive comments, indicating that the manuscript has substantially improved and is well written to be accepted.
